# Addressing Label Distribution Skew in Federated Learning with Per-Class Expert Models

## Abstract

Federated learning enables collaborative model training across distributed and privacy-sensitive datasets without sharing raw data. However, data heterogeneity, including label distribution skew, remains a major challenge, leading to performance degradation in highly non-iid settings. Despite extensive research, existing methods still struggle in severe settings. We explore a class-expert approach, where clients only contribute to the training of classes for which they have samples, thereby leveraging their specific expertise. Our current method employs per-class expert models trained in a simplified binary classification task, whose feature extractors are later concatenated to form a comprehensive global representation. Preliminary experiments show promising results in highly skewed scenarios, indicating the potential of class experts to improve performance under extreme non-iid conditions. Ongoing work focuses on enhancing scalability, computational efficiency, and stability.

## 1 Introduction

Machine learning and artificial intelligence rely on large volumes of high-quality data. However, collecting and centralizing such data is often infeasible due to privacy restrictions, legal limitations, and computational constraints [1, 2]. Sensitive information cannot easily be shared, and single institutions often lack sufficient samples for specific tasks. Moreover, centralized training demands substantial storage and processing resources [2].

Federated learning provides a decentralized alternative, enabling multiple data owners, called clients, to collaboratively train a shared global model without exposing their raw data [3]. In the standard FedAvg algorithm [4], clients perform local training and send model updates to a central server, which averages them to update the global model. This setup preserves privacy while leveraging distributed computation, and additional techniques such as differential privacy can further enhance data protection [1].

Despite these advantages, federated learning faces major challenges from data heterogeneity, since client data is typically not distributed independently and identically (non-iid) in real-life use cases [3, 2]. Variations in local data distributions can significantly degrade the global model's performance [5]. Among different types of non-iidness, including quantity, feature, and label skew, label distribution skew is particularly challenging learning bias. It is a prime example of class imbalance problems that affect model generalization and robustness. Zhao et al. [5] showed that FedAvg suffers from severe degradation under distribution skew due to client drift [6]. Clients optimize different local objectives [7], driving their models toward biased minima that favor majority classes and neglect minority or missing ones [8]. Aggregating such updates misaligns the global model and slows or even prevents convergence.

In this work, we propose leveraging multiple class-dependent expert models to improve feature extraction under label skew. Each expert is trained only on clients possessing relevant samples, mitigating missing-class effects and partially addressing local label imbalance. This specialization simplifies each model's classification task compared to a single global model. Preliminary results on CIFAR-10 are promising, showing improvements in highly skewed settings, though scalability and stability remain open challenges and the focus of ongoing research.

## 2   Related Works

Beyond the FedAvg algorithm [4], several approaches have been proposed to improve federated learning under heterogeneous data distributions. FedProx [9] introduces a proximal term to improve in heterogeneous settings. SCAFFOLD [10] further addresses client drift through control variates. More recent work has focused specifically on label distribution skew. FedLC [8] mitigates overconfidence on majority classes by calibrating model logits. FedCPD [11] leverages a pretrained vision transformer as a feature extractor and aggregates classifiers separately based on class availability. Contrastive learning is used to enhance feature alignment and representation robustness [12, 13]. Other approaches focus on feature extractor learning utilizing optimal simplex equiangular tight frame classifiers [14, 15], or employ sharing of not sensitive features to guide the global model while preserving data privacy [16, 17]. The following two methods are particularly relevant to our work.

FedConcat [18] mitigates label skew through model concatenation. Clients are grouped based on their class distributions to form clusters with similar label proportions. Within each cluster, a model is trained using FedAvg and cross-entropy loss. The feature extractors of all clusters are then concatenated into a single fixed feature extractor, and a classifier is trained on top. This concatenation reduces heterogeneity within clusters and enables the global model to learn complementary features from different data distributions. However, cluster imbalance and infrequent client participation can lead to undertrained cluster models, limiting performance in non-iid settings [19].

FedVLS [20] adapts the local loss function in the FedAvg algorithm to address label skew. The loss combines three terms: a calibrated cross-entropy term ($\mathcal{L}cal$) that reduces overconfidence in majority classes, a logit-based penalty ($\mathcal{L}logit$) emphasizing minority classes, and a distillation term ($\mathcal{L}_{dis}$) that preserves knowledge from the global model about missing classes: $\mathcal{L}_{VLS}(\omega) = \mathcal{L}_{cal}(\omega) + \lambda \mathcal{L}_{dis}(\omega) + \mathcal{L}_{logit}(\omega)$. For further details on the individual components, please refer to [20]. FedVLS achieves strong performance without additional communication overhead but still degrades under extreme label skew. This motivates its combinations with architectural strategies such as expert groups.

## 3   Methodology

**Preliminaries.**   In federated learning, $s$ clients with local datasets $\{D_i\}_{i=1}^{s}$ collaboratively train a shared global model. We consider a multiclass classification task with $K$ classes, where the local dataset of client $i$ is defined as $D_i = \{(x_j, y_j)\}_{j=1}^{N_i}$ and contains $N_i$ samples with input features $x_j$ and corresponding labels $y_j \in \{1, \ldots, K\}$. The objective is to learn model parameters $\omega$ that minimize the global loss function

$$\mathcal{L}(\omega) = \sum_{i=1}^{s} \frac{N_i}{N} \mathcal{L}_i(\omega),$$

where $N = \sum_{i=1}^{s} N_i$ is the total number of samples across all clients and the local loss at client $i$ is defined as $\mathcal{L}_i(\omega) = \mathbb{E}_{(x,y) \sim D_i} [\, l_i(f(x; \omega), y) \,]$. Here, $f(x; \omega)$ represents the model's prediction for input $x$, and $l_i(f(x; \omega), y)$ measures the discrepancy between the prediction and the true label $y$. Training proceeds over multiple communication rounds, where clients perform local updates and a central server aggregates the resulting models. [4].

**Label Distribution Skew.**   In practice, client data are rarely identically and independently distributed (iid) across decentralized clients [21]. Instead, clients often exhibit label distribution skew. Each client's data is sampled from a local distribution $(x, y) \sim P_i(x, y) = P_i(y|x)P_i(y)$ with

$$P_i(y) \neq P_j(y) \quad \text{but} \quad P_i(y|x) = P_j(y|x), \; \forall i \neq j.$$

This implies that each client observes different class frequencies. Some classes may be overrepresented (majority classes), underrepresented (minority classes), or entirely absent (missing classes). So, the distribution of classes can be unbalanced within a client and can vary from client to client. However, since $P_i(y|x) = P_j(y|x)$ is assumed, the input data $x$ is sampled from the same conditional distribution given the label, i.e., data characteristics per class remain consistent across clients [2], [8].

**Expert Models.** To address class imbalance and missing-class issues, we introduce a novel *expert group* approach. Clients are organized into $C$ expert groups with one group per class, and each group is responsible for training a model specialized in that class. Clients participate in an expert group if they possess at least one sample of the corresponding class, and may thus join multiple groups. Each expert model is trained over several communication rounds to jointly learn class-specific feature representations and a corresponding classifier.

In contrast to previous approaches, the classification objective in this phase is reformulated. Each expert model is trained as a binary classifier that distinguishes its target class from all others. This simplification reduces task complexity and improves training robustness. Because each participating client has samples of the target class, missing-class issues are eliminated. Additionally, aggregating all other classes into a single "others" category can further reduce local imbalance. The resulting expert models form a stacked feature extractor, where each component focuses on features distinctive to one class.

**Algorithm Overview.** Training proceeds in two stages: (1) expert model training and (2) global classifier construction. First, each expert model is trained federatively by clients that contain at least one sample of the corresponding class. At the beginning of each round, the server distributes the current expert models to eligible clients, which locally update them using the $\mathcal{L}_{VLS}$ loss function proposed in [20]. Since participating clients always possess samples of the expert class, and single-class clients are excluded , the term $\mathcal{L}_{dis}$ becomes zero and thus the loss function simplifies to:

$$\mathcal{L}'_{VLS} = \mathcal{L}_{cal} + \mathcal{L}_{logit}, \tag{1}$$

with

$$\mathcal{L}_{cal} = -\mathbb{E}_{(x,y) \sim D_i} \log \left( \frac{p(y)e^{f(x;\omega)[y]}}{\sum_c p(c)e^{f(x;\omega)[c]}} \right), \tag{2}$$

$$\mathcal{L}_{logit} = p(c) \log \left( \mathbb{E}_{(x,y) \sim D_i} \mathbb{I}(y \neq c)e^{f(x;w)[c]} \right), \tag{3}$$

where $p(y) = n_y/N_i$ denotes the empirical class frequency of label $y$ in the client's dataset $D_i$ and $f(x;w)[c]$ denotes the c-th entry of the model's prediction vector. The indicator function $\mathbb{I}(y \neq c)$ equals 1 if the current sample with label $y$ is misclassified as class $c$ and 0 otherwise. Although local imbalance may persist, this loss compensates for residual skew and corrects for overconfidence in majority classes and penalizes misclassification of minority class samples as majorities. Model updates are aggregated within each expert group using weighted averaging, proportional to the number of class samples per client, and thus the model weights for the next round, $\omega_{t+1}$, are given by:

$$\omega_{t+1} = \sum_{i=1}^{s} \frac{N_i}{N} \omega_t^i$$

At the end of the first stage, the feature extractors of all expert models are concatenated to form a stacked global representation. Each extractor captures features specific to its class, and their combination provides a rich, class-separating embedding for the full $C$-class task. Then, in the second stage, a global classifier is trained on top of the frozen stacked feature extractor using FedAvg and standard cross-entropy loss. Since the feature extractors already encode discriminative representations, the classifier only needs to learn decision boundaries, enabling efficient training without specialized loss functions.

# 4 Experiments

**Dataset.** We evaluate our method on the CIFAR-10 benchmark dataset [22] for 10-class image classification. The original train-test split is preserved, and the training set is further partitioned among 10 clients, each receiving approximately the same number of samples. Following prior work [20, 23], we simulate label skew using a Dirichlet distribution denoted as $\mathrm{Dir}(\beta)$. The parameter $\beta$ controls the degree of skewness, and we use $\beta \in \{0.05, 0.1, 0.5\}$.

128 **Baselines.**    We compare our approach with FedVLS [20] and FedConcat [18], which form the basis
129 for parts of our method. As FedVLS has been shown to outperform prior approaches, it is considered
130 the current state of the art and serves as the primary baseline. The reported results for both methods
131 are taken from [20].

132 **Implementation.**    To allow comparison, we closely follow the implementation details of FedVLS
133 [20]. All models use a ResNet-18 backbone [24] as feature extractor, and a single fully connected
134 layer as classifier. Training employs a batch size of 64 and uses the same data augmentation scheme
135 as FedVLS. Optimization is performed using stochastic gradient descent with a learning rate of 0.01,
136 momentum of 0.9, and weight decay of $10^{-5}$. Preliminary experiments indicate that 5 local epochs,
137 100 communication rounds in the first stage, and 20 in the second stage yield best results, matching
138 the settings in [20]. To assess robustness, each experiment is repeated five times.

139 **Performance comparison.**    Table 1 presents the results on CIFAR-10 under varying degrees of
140 label skew. For moderate skew ($\beta = 0.5$), all methods perform similarly, achieving accuracies above
141 90%. As the skew increases, our method exhibits reduced stability, particularly for $\beta = 0.05$, where
142 the standard deviation is notably high. Figure 1 illustrates the accuracy of the expert model approach
143 during the second training phase of the individual runs and their mean. While strong runs significantly
144 outperform the baselines, reaching accuracies above 80%, instability in other runs leads to a lower
145 overall mean, highlighting the need for improved robustness.

| Method | CIFAR-10 | | |
| --- | --- | --- | --- |
| | $\beta = 0.05$ | $\beta = 0.1$ | $\beta = 0.5$ |
| FedConcat | 64.30% ± 0.28 | 82.83% ± 0.21 | 92.45% ± 0.29 |
| FedVLS | 75.71% ± 0.28 | 84.35% ± 0.04 | 92.66% ± 0.14 |
| Ours | 71.41% ± 10.14 | 85.28% ± 3.04 | 92.45% ± 0.50 |

Table 1: Performance overview on CIFAR-10.

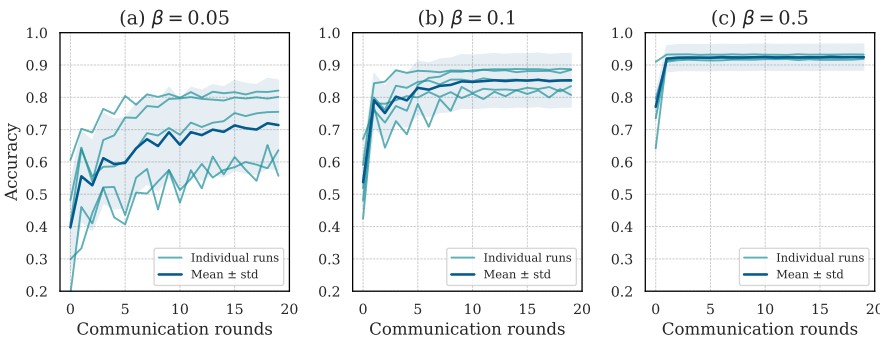

Figure 1: Accuracy of the expert models method during the second training phase over five independent runs and their mean.

146 **Expert model results.**    To better understand the limitations, we analyzed the accuracy of all expert
147 models during the first training phase, as shown in Figure 2. Evaluation was conducted on the global
148 test set using reformulated binary labels ("target class vs. all"). For moderate label skew, all expert
149 models achieve equal performance, with higher accuracy observed as the data becomes less skewed.
150 In the highly skewed setting ($\beta = 0.05$), some expert models maintain strong performance, while
151 others struggle significantly. This variation is attributed to random data partitioning, which results
152 in different degrees of heterogeneity between classes. Looking at the class distribution for this run
153 shows that, for the poorly performing class 6, a single client holds all but one sample. This highlights
154 the importance of further investigating such extreme cases to improve robustness.

## 5  Limitations and Outlook

156 The concept of class experts is part of ongoing research, and the proposed method is not yet practical.
157 The following limitations and potential directions represent the focus of our current work.

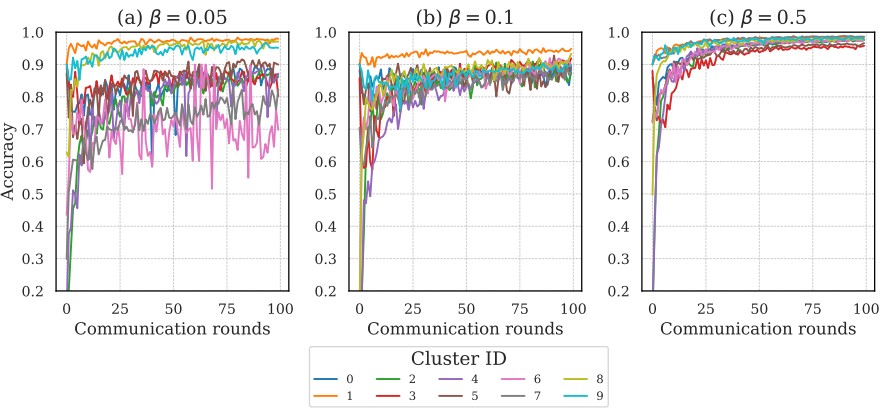

Figure 2: Test accuracy for binary tasks of individual expert models in the first training phase.

**Scalability.** In the current design, one model is trained per class, which does not scale to tasks with a large number of classes, as training hundreds of models becomes computationally infeasible. Preliminary experiments with multi-class expert groups (e.g., 2-vs-all or 3-vs-all) show potential for reducing the number of models, but selecting appropriate expert group compositions remains challenging compared to the simpler 1-vs-all setup. Ongoing work explores architectural modifications toward a shared feature extractor that preserves class-specific separation, allowing clients to contribute only to the relevant model components. Additionally, we aim to enable interaction between expert models during training to promote knowledge sharing across classes.

**Computational Costs.** As discussed above, the current expert model design is not computationally efficient. In settings with low label skew, where clients possess samples from most classes and therefore participate in many expert groups, the local computational cost increases substantially. Our current research focus on highly skewed scenarios, where improvements over existing methods are most needed. The scalability strategies outlined above are also expected to reduce computational overhead. Furthermore, we plan to investigate whether limiting the number of expert groups a client contributes to can maintain comparable performance while lowering the computational costs.

**Robustness.** Repeating the same experimental settings multiple times revealed robustness issues, particularly under the Dirichlet distribution with $\beta = 0.05$. A closer analysis of the data distributions showed that extreme class imbalances, which occasionally occur in this simulation setting, are a major contributing factor. As discussed in Section 4, classes with highly uneven distributions remain challenging. To address this, we are developing dynamic weighting strategies for model aggregation that prioritize reliable clients with more balanced data and sufficient target-class samples.

## 6 Conclusion

Label distribution skew is a major challenge in federated learning, representing a specific instance of the broader learning biases that arise from heterogeneous data distributions. We present on a class-guided training concept in which clients contribute only to models for classes they have expertise in. Our current approach employs class-specific expert models that are trained independently and then concatenated into a single global feature extractor. This architectural strategy, which treats data imbalance as a source of specialized knowledge, is a step towards developing more intrinsically robust machine learning systems.

Preliminary results are promising, particularly under highly skewed settings. While challenges in robustness, scalability, and computational efficiency remain open challenges and the focus of ongoing work, we believe this expert-based solution offers a generalizable path forward. It could be extended to mitigate other ML problems such as dataset shifts and misleading correlations.

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
