# OpenReview forum: "Addressing Label Distribution Skew in Federated Learning with Per-Class Expert Models"
_EurIPS.cc/2025/Workshop/UPLB — UPLB2025_

### Official Review · Reviewer_a989 · 2025-10-17
**Interesting idea, very preliminary results**

**Rating:** 5
**Confidence:** 3

**Review:**

The paper proposes a class-expert approach to mitigate label distribution skew in federated learning. Each client contributes only to models corresponding to classes present in its local dataset, leading to per-class expert models that are later concatenated into a global representation. Preliminary results on CIFAR10 are interesting under highly non-IID settings, though robustness and scalability remain open issues.

Strengths:
- Novel architectural perspective: The idea of treating label imbalance as a form of class expertise is original and conceptually appealing. It reframes heterogeneity from a nuisance into a potential source of specialization.
- Two-stage training process is clearly described and easy to follow.
- Experiments seem aligned with prior baselines (FedVLS, FedConcat), although I'm not an expert in this area.

Drawbacks
- Limited empirical validation: results are restricted to CIFAR10 in small-scale settings, and no experiments on more realistic federated environments are presented. I think it's still acceptable for a workshop.
- Scalability concerns (mentioned by the authors): Training one model per class is not practical for problems with many classes. The proposed multi-class grouping extensions are speculative.
- The approach lacks a theoretical analysis linking the class-expert decomposition to global convergence or fairness properties.

This is an interesting early-stage contribution, but the paper would benefit from stronger analysis, larger-scale experiments, and explicit connections to fairness and distributional robustness literature. The fit for the workshop is adequate.

---

### Decision · Program_Chairs · 2025-11-03

Accept (Poster)